# Application of the “SCOBY” and Kombucha Tea for the Production of Fermented Milk Drinks

**DOI:** 10.3390/microorganisms9010123

**Published:** 2021-01-07

**Authors:** Marcin Kruk, Monika Trząskowska, Iwona Ścibisz, Patryk Pokorski

**Affiliations:** 1Faculty of Human Nutrition, Warsaw University of Life Sciences—SGGW, Nowoursynowska 159c, 02-776 Warsaw, Poland; s187662@sggw.edu.pl (M.K.); s197110@sggw.edu.pl (P.P.); 2Chair of Food Hygiene and Quality Management, Department of Food Gastronomy and Food Hygiene, Institute of Human Nutrition Sciences, Warsaw University of Life Sciences—SGGW, Nowoursynowska 159c, 02-776 Warsaw, Poland; 3Department of Food Technology and Assessment, Division of Fruit and Vegetable Technology, Institute of Food Sciences, Warsaw University of Life Sciences—SGGW, Nowoursynowska 159c, 02-776 Warsaw, Poland; iwona_scibisz@sggw.edu.pl

**Keywords:** fermentation, food processing, functional food, kombucha, milk, symbiotic culture of bacteria and yeast (SCOBY)

## Abstract

For the production of fermented milk drinks, cultures of microorganisms other than traditionally applied can be used. Such possibilities are created by the symbiotic culture of bacteria and yeast (SCOBY), which is used to produce kombucha. The aim of the study was to evaluate the possibility of using kombucha and the SCOBY for fermented milk drink products. The drinks were developed with a lactose-free variant and traditional milk. For the analysis of the obtained beverages, microbiological methods (CFU method), chemical methods (pH method and HPLC method) and the quantitative descriptive analysis (QDA) sensory method were used. As a result of the research, a recipe and the fermentation parameters for fermented milk drinks were developed. In the developed lactose milk drinks, the average lactose content was 4.25 g/100 g. In lactose-free milk drinks, the average glucose content was 2.26 g/100 g. Lactic acid in both types of products was at the highest average level of 0.68 g/100 g. The products had a characteristic pH value for fermented milk drinks and a very good microbiological quality, which followed the FAO/WHO guidelines. Drinks also had a typical sensory profile for this products group. However, slight sensory defects were detected. The developed fermented milk drinks have a potential health-promoting value, thanks to the content of active microflora and organic acids, which have a confirmed positive effect on the human body. The drinks produced require further testing to optimize their cost of production, possible health benefits and sensory quality.

## 1. Introduction

In the last decade, the expansion of pro-health trends in food production and consumption has been observed. One of them is the development of functional food, which thanks to specific properties can contribute to reducing the occurrence of diseases and improve the quality of life [1,2]. This type of food contains products from different groups, including various types of dairy products [3,4]. In addition, food disorders like lactose intolerance are becoming an increasing nutritional problem. This stimulates the growth of the lactose-free fermented milk drinks market and the design of new products.

Fermented milk drinks contain many substances that have a positive effect on the human body [5]. The chemical composition of milk during fermentation is changed by certain microorganisms. Namely, three types of processes are most often observed, i.e., lactic acid fermentation, alcoholic fermentation caused by yeast and oxidation of carbohydrates and alcohols caused by acetic acid bacteria (AAB), which is commonly called acetic acid fermentation [6]. Thanks to the active microflora, proteins, fats and carbohydrates are transformed, which compose the basic nutritional value of fermented milk [5]. Differences in the bioavailability of nutrients in fermented milk drinks compared to sweet milk were also observed. For example, the better bioavailability of calcium, protein and carbohydrates was observed [7]. A large group of compounds that are formed during fermentation are metabolites of the fermentation microflora. Many of them have health-promoting features that shape the nutritional value of this group of food products. These can be vitamins, organic acids, exopolysaccharides such as kefiran or polypeptides [8].

An aspect of great importance due to the functional properties of fermented milk beverages is the content of active probiotic microflora. Food products with the addition of probiotic microorganisms are currently the leading consumption trend [9]. According to the FAO and WHO [10], probiotics are “live microorganisms that, when administered in appropriate amounts, confer health benefits to the host.” Strains of microorganisms that show probiotic effects are most often identified from the lactic acid bacteria (LAB) genera such as *Lactobacillus, Lactococcus*, *Streptococcus*, *Oenococcus*, *Pediococcus*, *Leuconostoc*, *Enterococcus* and *Bifidobacterium*. Some yeast strains may also be included in the probiotic microorganisms, in particular *Saccharomyces cerevisiae*, as well as some mould strains including *Aspergillus oryzae* and *A*. *niger* [10,11]. The most important mechanisms of the impact of probiotics on the human body include the modulation of the immune system, the synthesis of substances antagonistic to pathogenic and putrefactive bacteria, the competition for nutrients with pathogenic and putrefactive microorganisms, blocking the colonization of the digestive system by pathogens as a result of adhesion to the intestinal walls, the assimilation and inhibition of the synthesis of toxins by pathogenic bacteria, as well as anti-tumour activity consisting of inhibiting carcinogenic compounds [12,13,14,15,16]. Due to the growing awareness of the health-promoting effect of probiotics, researchers are looking for other groups of microorganisms than those currently known and described as probiotics. Neffe-Skocińska [17] and Haghshenas [18] identified AAB strains and conducted basic tests on their probiotic properties, for which they obtained positive results.

The assortment of fermented milk drinks on the market includes mainly yoghurts, kefirs, acidophilic milk, buttermilk and koumiss [19]. This presents a potential application for the fermentation of milk by the SCOBY (symbiotic culture of bacteria and yeast). The SCOBY is also called the tea mushroom because of its similar shape and appearance to the fruiting caps of macroscopic mushrooms. It is a cellulosic biofilm formed by the polymerization of monosaccharides. The SCOBY microflora includes AAB, yeast, LAB and bifidobacteria [20]. The tea mushroom is used as a starter culture for the production of the kombucha tea drink. It is a drink resulting from the fermentation of a sweetened tea infusion by SCOBY, and it is described as “slightly sweet, slightly sour refreshing drink” [20,21,22,23,24]. According to the current state of knowledge, SCOBY and kombucha tea drinks do not have confirmed health-promoting properties for humans. However, there is a large database of results demonstrating the potential wholesome properties that have been obtained through in vitro and in vivo animal studies [25]. Kombucha tea drinks contain several main groups of bioactive compounds that are metabolized by the SCOBY. These include organic acids, water-soluble vitamins and polyphenolic compounds [20,26]. A great chance to identify potentially probiotic bacteria from the SCOBY is due to the content of various groups of microorganisms, including LAB and bifidobacteria, which include the largest group of proven probiotic strains [27]. Thanks to its specific fermentation properties, the SCOBY has been used in the production of food products other than kombucha tea drinks [28,29,30]. One of the directions of the SCOBY’s usage may be the fermentation of milk. Attempts were made to develop fermented beverages based on milk and the tea mushroom, although they did not have the appropriate sensory features [31,32,33].

The aim of the study was to assess new technological solutions for the use of the “SCOBY” and kombucha tea for the production of fermented milk drinks that will result in a very good sensory, microbiological and chemical quality of the obtained fermented product. The applied research methods allow for comprehensive determination of the quality of the developed products. This knowledge is crucial for further, more detailed research of the product microflora and its processing.

## 2. Materials and Methods

A diagram of the experiment, as well as variations of the examined samples of milk inoculum and fermented milk beverages are presented in Figure 1.

### 2.1. Materials

#### 2.1.1. Kombucha Tea

Kombucha tea was prepared from 1 litre of tap water, 2 g of green and 4 g of black leaf tea (Lord Nelson, Lidl; Warsaw Poland), which was brewed for 15 min at 90 °C. Sucrose was added to the tea infusion (Diament, Poland) in the amount of 100 g/L. When the tea infusion was cooled to 22 °C, the SCOBY starter culture weighing 50 g/L and the kombucha tea (Fermentaholics, Clearwater, FL, USA) in an amount of 60 mL/L were added. Fermentation was carried out for 10 days at 22 °C. After this time, the SCOBY biofilm and kombucha tea drink were obtained and were used to produce the inoculum of the fermented milk beverages.

#### 2.1.2. Milk Inoculum (Lactose and Lactose-Free)

Milk inoculum was prepared in two variants: from traditional milk with lactose (LM; Łaciate, Mlekpol, Poland) and lactose-free milk (LFM; Wydojone, Mlekovita, Poland). Each variant consisted of UHT milk (1.5% fat) in the amount of 78%, SCOBY 10%, kombucha tea infusion 8% and powdered skim milk for a total of 4% (Mlekovita, Poland). Milk and milk powder in both versions were mixed, homogenized and then pasteurized at 90 °C for 5 min. After cooling to 42 °C, the prepared base was inoculated with the SCOBY and kombucha tea. Fermentation was carried out for 12 or 24 h at 42 °C.

#### 2.1.3. Fermented Milk Beverage (Lactose and Lactose-Free)

Fermented milk beverages were prepared in two variants: from traditional milk with lactose (LM; Łaciate, Mlekpol, Grajewo, Poland) and lactose-free milk (LFM; Wydojone, Mlekovita, Wysokie Mazowieckie, Poland). Milk and milk powder in both versions were mixed in the proportion of 24:1, pasteurized at 90 °C for 5 min and then cooled to 42 °C. The milk inoculum was introduced into the prepared mixtures in the amount of 2.5% and 5%. Fermentation was carried out at 42 °C until the milk was fully coagulated; this was 7–8 h.

### 2.2. Methods

#### 2.2.1. Sensory Analysis

For the evaluation of the sensory quality of the fermented milk beverage variants, the quantitative descriptive analysis (QDA) [34] method was applied. The analysis was carried out by a group of fourteen experts using an unstructured linear scale (0–10 conventional units (c.u.)). Firstly, a discussion among experts was performed, after which the discriminants of the fermented milk beverages were defined. During analysis, the following characteristics were assessed: the intensity of milk, boiled milk, sour, sweet, yoghurt, caramel, rancid and other odours. Furthermore, smoothness, thickness and syneresis, as well as the intensity of milk and boiled milk, sour, bitter, yoghurt caramel, rancid and other flavours together with the overall quality were evaluated. Overall quality in the QDA method is expressed as the harmonization of all product characteristics and is not related to the expression of the hedonic quality. For the discriminants related to the intensity of the perception of a given odour or flavour, the boundary terms were “none” and “very strong”. The terms of product consistency were matched to the perceived characteristics of a given product feature. Boundary terms related to the overall quality of products were “low” and “high”. The samples for analysis were placed in transparent plastic containers with a cover, in a volume of 25 mL. The samples were coded with random numbers and given in random order. To neutralize the taste, boiled water at room temperature was served between the samples.

#### 2.2.2. Microbiological Analysis

The number of lactic acid bacteria (LAB), acetic acid bacteria (AAB), bifidobacteria (BB) and yeasts were determined in all variants of inoculum and fermented beverages. One millilitre of the tested sample was taken, and nine millilitres of sterile peptone water (Biocorp, Poland) were added, then decimal dilutions were made. The number of lactic acid bacteria was determined by the pour plate technique. The petri dishes with the appropriate decimal dilution had MRS agar (Biokar Diagnostic, France) poured in them, and they were incubated at 37 °C for 48 h. The number of acetic acid bacteria was determined by the spread plate technique. The medium consisted of 50 mg/L natamycin (Sigma-Aldrich, Poland), 0.3% casein peptone (Merck, Poland), 0.3% yeast extract (Merck, Poland), 0.7% calcium carbonate (Sigma- Aldrich, Poland), 2% glucose (Sigma-Aldrich, Poland), 2% agar (Sigma-Aldrich, Poland) and 2% ethanol 96% (Poch, Poland). The petri plates with the solidified medium and the appropriate decimal dilution of the sample were incubated at 28 °C for 72 h. The number of bifidobacteria was determined by the overlay pour plate method. BSM agar medium (Sigma-Aldrich, Poland) was used with the addition of a BSM supplement in the amount of 116 mg/L (Sigma-Aldrich, Poland). Petri dishes with the solidified medium and the appropriate decimal dilution of the sample were flooded with an additional layer of BSM agar to limit the exposure to oxygen. The samples were incubated at 37 °C for 48 h. The number of yeasts was determined by the spread plate technique, and YGC agar (Biokar Diagnostic, France) was used. Appropriate decimal dilutions of the sample were applied to the petri dishes with the solidified medium, spread with a sterile spreader and incubated for 5 days at 25 °C. After incubation, typical colonies were counted. The tests were made in 3 replications.

#### 2.2.3. pH Analysis

The pH measurement was determined using a pehameter CP501 (Elmetron, Poland) with a precision of up to 0.01 according to the device’s manual. The tests of the milk inoculum and fermented milk beverages’ pH were performed in 3 replications.

#### 2.2.4. Chemicals

L-ascorbic acid, dithiothreitol (DTT), glucuronic acid, pyruvic acid, orotic acid, phosphoric acid, lactose, glucose and galactose acid were obtained from Sigma-Aldrich with purities of 97–99%. Acetic acid, citric acid and L(+)-lactic acid were purchased from Merck KGAA (Darmstadt, Germany). Meta-phosphoric, methanol and acetonitrile were obtained from Honeywell Fluka (Charlotte, NC, USA). All solvents used were HPLC grade. HPLC grade water (18 MW) was prepared using a Millipore purification system (Millipore, Germany).

HPLC analysis of sugars

Sugar content was determined by high-performance liquid chromatography according to the modified method of Zhang et al. [35] and Usenik et al. [36]. The samples of 1 g of fermented milk drinks or non-fermented milk were extracted with 9 mL of distilled water for 30 min at 25 °C with frequent stirring at 250 rmp (Unithermix, LLG Labware). After extraction, the samples were centrifuged for 10 min at 15,133× *g* at 4 °C (MPW-350R, MPW Med. Instruments, Warsaw, Poland). The supernatants were filtered through nylon syringe filters of 0.45 µm, transferred into vials and used for the analyses. Each sample was extracted in triplicate before HPLC analysis. The Shimadzu Prominence HPLC system equipped with an LC-20AD pump, a refractive index detector (RID-10A), an autosampler (SIL-20A HT), a column oven CTO-10ASVP, a degasser Model DGU-20A5R and LCsolution data collection software was used to analyse the sugars. Separation was carried out using the RezexTM RCM-Monosaccharide Ca+ column (300 × 7.8 mm) (Phenomenex, Torrance, CA, USA) with the column temperature maintained at 65 °C. The mobile phase used was Milli-Q water eluted on an isocratic gradient with a flow rate of 0.6 mL × min^−1^. The injection amount was 20 µL. Lactose, glucose and galactose were identified and quantified by the external standard method, and the content was expressed as g per 100 g of fermented milk drinks or non-fermented milk. The correlation coefficient obtained from the standard sugar calibration curves was up to 0.996.

HPLC analysis of organic acids

The analysis of organic acids in fermented milk drinks or non-fermented milk was performed according to the procedure described by Fernandez-Garcia and McGregor [37] and Ścibisz et al. [38] with minor modification. Two grams of sample were diluted to 25 mL with distilled water. Then, the solution was stirred in a shaking thermostat with a constant speed of 250 rmp (Unithermix, LLG Labware). After 30 min of shaking, the extracts were centrifuged for 10 min at 15,133× *g* at 4 °C to separate the layers. Finally, the supernatants were filtered through a 0.45 µm pore size membrane filter and transferred into a vial. The analysis was performed using a Shimadzu HPLC system with a diode array detector (DAD) (SPD M20A). Chromatographic separation was carried out isocratically at 22 °C with a mobile phase of 20 mmol phosphoric acid at a flow rate of 0.7 mL × min^−1^ on the Cosmosil 5C18-PAQ (4.6 mm × 150 mm) column (Waters, Etten-Leur, The Netherlands). A wavelength of 210 or 254 nm was used for the detection, and the individual organic acids (lactic, citric, acetic, glucuronic, pyruvic, orotic) in fermented milk drinks or non-fermented milk were identified by their spectral and retention time characteristic. Analysis was done in triplicate for each sample, and the content of organic acid was expressed as g per 100 g of fermented milk drinks or non-fermented milk.

Vitamin C determination

Vitamin C concentration, as the sum of ascorbic acid and dehydroascorbic acid, was determined by the method proposed by Chebrolu et al. [39] with slight modifications. For quantification of total vitamin C, dehydroascorbic acid was reduced to ascorbic acid by the addition of dithiothreitol (DTT). Vitamin C was extracted from fermented milk drinks or non-fermented milk using the protocol described by Chotyakul et al. [40]. Briefly, five grams of fermented milk drinks or non-fermented milk were placed in a volumetric flask and diluted to 10 mL with a solution of 2% metaphosphoric acid. The solution was transferred to the centrifuge tubes, mixed with a vortex at high speed for 10 s and centrifuged for 10 min at 15,133× *g* at 4 °C. After filtration (syringe filters of 0.45 µm), the supernatant (300 µL) was mixed with 300 µL 2% metaphosphoric acid and immediately injected into HPLC column for ascorbic acid analysis. The supernatant (300 µL) was also mixed with a solution of 10 mmol/L DTT, and after 30 min, the sample was injected into the HPLC system with a DAD (SPD M20A). Chromatographic separation was achieved using an Onyx Monolithic column (100 × 4.6 mm, Phenomenex, Torrance, CA, USA) with guard cartridge Onyx Monolithic C18 (10 × 4.6 mm, Phenomenex, Torrance, CA, USA). Zero-point-one percent phosphoric acid was used as the mobile phase at a flow rate of 1 mL × min^−1^. The injection amount was 20 µL, and the ascorbic acid peak was detected at 254 nm. All samples were prepared in triplicate. For quantification purposes, a standard calibration curve was obtained by plotting peak areas vs. the series concentration of L-ascorbic acid.

#### 2.2.5. Statistical Analysis

The statistical analysis of the results was performed using the Statistica 13.3 software (StatSoft, Kraków, Poland). The arithmetic mean and standard deviation were calculated. Moreover, the multivariate analysis of variance (ANOVA) was used, and the results of the sensory analysis were subjected to principal components analysis (PCA). Fisher’s NIR test was used to compare post-hoc means. The difference was considered statistically significant when *p* < 0.05 with regard to the number of microorganisms, pH value, content of organic acids and vitamins and the results of sensory evaluation.

## 3. Results

### 3.1. Inoculum and Fermented Milk Drinks’ Microbial Characteristics and pH Value

Table 1 presents the results of the microbiological evaluation of the designed inoculum. The content of LAB in inoculums was on average 9 log CFU mL^−1^. The BB number was an average of 7 log CFU mL^−1^. The type of milk and the fermentation time did not significantly influence the LAB and BB number (*p* > 0.05). For AAB, a significant influence of time fermentation was observed between the LM12 and LM24 samples (*p* < 0.05). All other samples regarding AAB content did not differ significantly among themselves. The number of yeasts was affected by fermentation time and the type of milk. The differences were observed between both samples from lactose milk and the sample fermented by 24 h from lactose-free milk (*p* < 0.05). In general, the 12 h fermented inoculum for both types of milk had better microbiological quality.

The type of milk and fermentation time had a significant impact on the pH parameter (*p* < 0.05). Samples fermented for 24 h had a lower pH than those fermented for 12 h. All samples were statistically different from each other (*p* < 0.05) except samples fermented for 12 h (*p* > 0.05) (Table 1).

Table 2 presents the results of the microbiological evaluation of the designed fermented milk beverages. The LAB count was near 9 log CFU mL^−1^ in all variants of the fermented beverages and did not differ significantly (*p* > 0.05). Another situation could be seen in the case of BB, and the effect of the type of milk on the number of bacteria was observed (*p* < 0.05). The highest number of the BB was in the LFM-12-2.5 sample, which was significantly different from all lactose milk samples (*p* < 0.05). Furthermore, differences in the number of BB were noted in the LFM-24-2.5 and LM-12-5, as well as LFM-12-5 samples (*p* < 0.05). The content of AAB in the fermented milk beverages was on average 9 log CFU mL^−1^ and influenced by milk type and fermentation time (*p* < 0.05). Sample LM-24-2.5 was substantially different from samples LM-12-2.5, LM-12-5, and LFM-24-5 (*p* < 0.05). The others did not differ in AAB content (*p* > 0.05). The average number of yeasts in products was 5.23 CFU mL^−1^. The number of yeasts in products was diversified. The highest count of yeasts was in sample LM-12-2.5 and the lowest in LM-24-2.5. However, there were no significant differences among the analysed samples (*p* > 0.05).

The type of milk did not significantly influence the pH value (*p* > 0.05), but the fermentation time and inoculum content had a significant impact on the pH value of the fermented milk beverages. The samples fermented with a 24-h inoculum had a higher pH value than samples with a 12-h inoculum. Samples with a 5% addition of inoculum fermented for 12 h had the lowest pH value. These samples were statistically different from the others (*p* < 0.05) (Table 2).

### 3.2. Determination of Organic Acids and Carbohydrate

The results of the chromatographic analysis are presented in Table 3. Lactose was not detected in the LFM samples. In the case of the LM samples, significant differences in the content of this disaccharide occurred between the samples with the 12 and 24-h inoculum (*p* < 0.05). The lactose content of all fermented samples was significantly lower than the unfermented standard (*p* < 0.05). The variability of the glucose content in the LFM samples was similar to the variability of the lactose content in the LM samples. The fermented products with content of different inoculum fermentation times were significantly different from each other (*p* < 0.05). With regard to galactose, significantly less of this sugar was present in the samples with the 24-h inoculum than in those with the 12-h inoculum (*p* < 0.05). In the LM samples, the galactose content increased in the samples fermented with the 24-h inoculum (*p* < 0.05). A similar relationship was observed in samples fermented with the 12-h inoculum compared to LM-C (*p* < 0.05). Vitamin C content did not differ significantly between the analysed samples (*p* > 0.05). The products were also analysed for the presence of acetic acid, but it was not detected. Among all the analysed organic acids, lactic acid had the highest share. The LM samples differed significantly in the content of this acid between those with the 12-h inoculum and the 24-h inoculum (*p* < 0.05). For the LFM samples, the differentiation factor was also the fermentation time of the inoculum used to start the milk fermentation (*p* < 0.05). All fermented products differed significantly from the controls (LM-C; LFM-C) (*p* < 0.05). In relation to citric and orotic acid, there were no substantial differences within the product group of LM and LFM. They also did not differ from the control samples, which means that these components of the product composition did not change during fermentation. Glucuronic acid was metabolized during milk fermentation. Its presence was not detected in unfermented controls. The differences found between the samples were significant in all cases (*p* < 0.05), although the content of this acid in individual samples did not differ more than 0.01 g. When it comes to pyruvic acid, statistical differences were found only between non-fermented control samples and fermented samples (*p* < 0.05). These differences were found both in the LM and LFM groups. Additionally, it was noticed that the content of this acid decreased after milk fermentation.

### 3.3. Sensory Analyses

Figure 2 and Figure 3 show the results of the sensory evaluation of the developed product. ANOVA analysis provided information on the essential differences between samples and the effect of milk type, inoculum fermentation time and inoculum concentration on the sensory characteristics. No statistically significant differences were observed between the samples for the following features: milk, sour, yoghurt, rancid and other odour, smoothness, syneresis, milk, bitter, rancid, other flavour and overall quality (*p* > 0.05). The factor that most often differentiated samples was the type of milk. The significant differences between the samples were observed for boiled milk and caramel odour, as well as boiled milk, sweet, yoghurt and caramel flavour (*p* < 0.05). In the case of a sensory attribute of boiled milk odour, sample LFM-12-5 was different from all LM samples (*p* < 0.05). Statistical analysis also showed the impact of the type of milk (LFM) on the boiled milk flavour. The odour and flavour of caramel discriminators were significantly more palpable in LFM than in LM samples (*p* < 0.05). Another relation between the samples was observed in terms of sweet odour and flavour. The odour of fermented LM beverages differed significantly from LFM-12-2.5 and LFM-12-5 (*p* < 0.05), and the effect of fermentation time and inoculum content also was noted. The influence of these factors was observed between LFM-12-5 and LFM-24-2.5 (*p* < 0.05). Considering the sweet flavour, the differences were between LM-12 and all LFM samples (*p* < 0.05). There were also significant differences between the LFM-12-2.5 and LM-24-2.5 and LM-24-5 drinks (*p* < 0.05). The influence of the type of milk on the yoghurt flavour was observed only between a few samples, i.e., LM-12-5, LFM-12-5, LM-24-2.5 and LFM-12-2.5 (*p* < 0.05). The sensory features that were affected by all analysed variables were sour flavour and thickness. Within the sour taste, the LM-12-5 sample differed from all others (*p* < 0.05). This sample also had the lowest pH value (Table 2), which could result in a given relation. In addition, LM-24-2.5 was different from LFM-12-2.5 and LFM-24-5 in terms of more intense sour flavour. The last feature in which statistical differences were shown was thickness. Within this texture attribute, differences were found between sample LM-24-5 and all others (*p* < 0.05) except LM-24-2.5 and LFM-24-2.5 (*p* > 0.05). The LM-24-5 sample was thinnest among the others. Additionally, the LFM-24-2.5, LM-24-5 and LM-12-2.5 samples did not differ significantly from LM-24-2.5 (*p* > 0.05), which was the thickest among the LM products. There were no differences between the analysed products in terms of overall quality (*p* > 0.05). This attribute is one of the most important features affecting most sensory features of products. To research these relations, PCA was performed. This analysis aimed to group sensory discriminants and describe the correlations among them. Nine principal components were identified for both groups of samples, which defined over 80% of the sensory attributes of the products. Samples variability was defined in 38% by the first and second principal components of LM. Within the discussed product group, the highest positive correlation with overall quality had the following sensory attributes: milk odour and flavour, as well as yoghurt odour and flavour; whereas the negative correlation with the overall quality was characterized by bitter flavour, rancid odour and flavour, other flavour and syneresis. In the case of LFM samples, a positive correlation was found between overall quality for the milk odour and flavour odour and smoothness. In contrast, a negative correlation occurred for the following sensory discriminants: bitter flavour, rancid odour, rancid flavour, other flavour and syneresis. Comparing the results of PCA and ANOVA, it was noted that most sensory characteristics correlated with the overall quality attribute did not differ significantly (*p* > 0.05). For this reason, there were no differences between the samples in terms of overall quality.

## 4. Discussion

Many factors influence the quality of fermented milk beverages. The determinant of the proper course of fermentation is the appropriate microbiological quality of the starting inoculum. Literature data indicate that the number of microorganisms in fermentation starters was at different levels. The number of LAB and yeast in the starters used in the commercial production of yoghurts and kefirs is similar to the results obtained. However, the number of AAB in starters used for kefir production and kombucha-based milk drinks was lower than in our research [32,41,42,43,44]. Regarding literature data, the obtained results of our research indicate a high content of active fermentation microflora in the developed variants of the inoculum (Table 1). One can also observe a change in the percentage of the examined microorganisms in comparison to their content in the SCOBY and kombucha, which had no contact with the milk environment. According to Marsh et al. [45] and Jayabalan et al. [20] in the composition of kombucha tea drinks, the dominant microorganisms are AAB, which most often constitute over 90% of the total number of microorganisms present in kombucha drinks and the SCOBY. The rest of the microorganisms were yeasts, LAB and also BB and other types of bacteria, which constitute a negligible percentage of the total number of the SCOBY and kombucha microorganisms. The results of their research indicate that the share of individual groups of microorganisms in the milieu of milk after fermentation has changed. The percentage of AAB decreased and the percentage of LAB, BB and yeast increased in relation to the total number of tested microorganisms (Table 2). This statement is consistent with the results obtained by Kruk et al. [31], who documented that the SCOBY and kombucha microorganisms could adapt to the environmental conditions of milk.

The obtained pH values were reflected in the literature data. Kruk et al. [31] investigated the pH value of the inoculum of fermented milk beverages. The results obtained by them depended on the fermentation time and were in the range of pH 4.5–4.7. Iličić et al. [46] fermented milk using the SCOBY. The pH value obtained after 12 h of fermentation was close to pH 4.5. Furthermore, Kruk et al. [31] studied the dynamics of pH value changes during milk fermentation using the SCOBY, and the results they obtained were similar to the presented research. However, the literature does not indicate unequivocal pH values that the mother culture used for milk inoculation should have. An important aspect is that in the studies presented in the literature, the SCOBY came from various sources, so the origin of the starter culture and its microbiological composition have a large influence on the pH values due to the fermentation time [45].

Guidance on LAB, BB and yeast content in fermented milk beverages is provided in the Codex Alimentarius. The number of active microflora in fermented milk drinks should not be less than 7 log CFU mL^−1^ of the product regarding LAB, 6 log CFU mL^−1^ of the product in products with the addition of probiotic bacteria such as BB and 4 log CFU mL^−1^ of yeast [19]. The number of these microorganisms in the developed milk drinks meets the requirements dictated by the FAO and WHO. Additionally, the developed drinks had a greater number of microorganisms than the one presented in the FAO and WHO guidelines, which means that all products were of very good microbiological quality. AAB are not a typical group of fermented milk microbes. However, they constitute the share of microflora included in kefirs and other regional milk-based beverages [8,47,48]. Some studies showed that the number of AAB in kefir was at the level of 5 log CFU mL^−1^ [49]. However, there are no regulations that define their content in fermented milk beverages. De Filippis et al. [50] and Neffe-Skocińska et al. [51] documented that in kombucha tea drinks, the AAB number was 7–8 log CFU mL^−1^ of the product. Kruk et al. [31] presented the results in which the AAB content in milk beverages fermented with the SCOBY was at the level of 8–9 log CFU mL^−1^ of the tested products. These values are reflected in our research and provide the conclusion that the milk environment and the established fermentation conditions may have a positive effect on the growth of AAB.

Despite the compliance of the results with the literature data, it was difficult to define the affiliation of the developed products to a specific group of fermented milk beverages, so it is not possible to indicate the pH value they should have according to the commodity characteristics of the products. The pH values of fermented milk beverages available on the market were in the range of 3.9–4.7. This value depends on the type of fermented milk drink, the time and the parameters of product fermentation [7,52]. The obtained test results are reflected in the literature data and proved the appropriate physicochemical quality of the products (Table 1 and Table 2).

Many authors conducted research on the possibility of using the SCOBY and kombucha for the production of fermented milk beverages, but the products they developed were not subjected to sensory evaluation or did not have appropriate quality characteristics for this assortment group [31,32,33,44,53,54]. Fermented milk drinks should have a set of flavour characteristics for this type of product. According to Karagül-Yüceer and Drake [55], fermented milk drinks should have a dairy, refreshing, sour, yoghurt-fermented aroma and taste. On the other hand, the consistency of fermented milk beverages should be smooth, dense or thin, depending on the type of drink [48,55,56,57]. The results of the presented research showed that all the developed variants of milk drinks had these quality features. Additionally, they were positively correlated with the overall quality of the tested products. There were also features that were highly related to sensory defects in fermented milk beverages, such as the odour and flavour of boiled milk, rancid and bitter, the presence of syneresis and a sandiness [55]. Among the analysed product characteristics, special attention should be paid to the odour and flavour of boiled milk, which achieves the highest values (3.6–5.5 c.u.) among the examined attributes responsible for the sensory defects of products. This defect could be caused by a too long pasteurization of milk [58]. The remaining determinants, such as the intensity of the rancid odour and flavour, the intensity of the bitter flavour and the range of syneresis, which define product defects, remained at a low level. An important aspect was their negative correlation with the overall quality discriminant. Sources presenting data on the production technology of fermented milk beverages state that the addition of the inoculum in the form of a mother culture should be between 2% and 5% of the mass of fermented milk to obtain the best physicochemical, microbiological and sensory characteristics of the finished product [59]. Additionally, Özer and Kirmaci [58] reported that when a lower or higher concentration of the inoculum for milk fermentation than those indicated above is used, flavour defects arise in the products.

The content of organic acids and sugars in the developed beverages is consistent with the data available in the literature. The sugar content in fermented milk beverages such as kefir or yoghurt is reduced by an average of 30% compared to their content in non-fermented milk [52]. Additionally, Vukic et al. [53] found that the reduction of sugar content in milk drinks fermented with kombucha is the same as in drinks available on the market. The obtained results confirm this relationship (Table 3). The variability in the content of galactose in the developed products was a completely standard phenomenon caused by the metabolic activity of the fermentative microflora [60]. The content of vitamin C in the fermented products did not change compared to that in the control samples. This result proves that AAB did not synthesize this vitamin. Variations in citric, orotic and pyruvic acid content are typical of fermented milk beverages [61]. These are acids whose content does not change significantly from their content in non-fermented milk [62]. The most characteristic organic acid of fermented milk beverages is lactic acid, the content of which in this group of products varies considerably, but most often, it is present in a range of 0.1 to 1 g/100 g of the product [52,53,61,63]. Its concentration in the prepared drinks oscillated around 0.58–0.77 g/100 g. These data are in line with the information available in the literature regarding lactic acid in kefirs, yoghurts and milk drinks fermented with kombucha. Glucuronic acid is a typical metabolite of acetic acid bacteria. It is produced by the oxidation of glucose [6,64]. Its presence in the analysed products was found only in fermented samples. Glucuronic acid was not detected in unfermented controls. This proves that this component of the products was created as a result of the metabolic activity of AAB. The obtained results are similar to the findings presented by other authors with regard to the content of glucuronic acid in the developed beverages [65,66]. Additionally, no acetic acid content was found in the tested samples. It is well known that acetic acid is produced by the oxidation of ethyl alcohol by AAB. Despite the large number of acetic acid bacteria in the samples, acetic acid was not formed due to the lack of a substrate for the bioreaction of the synthesis, which is ethyl alcohol. The absence of this acid proves the absence of ethyl alcohol in the tested samples or the metabolic inability of AAB strains to synthesize this component Additionally, the synthesis of organic acids by AAB is a species-dependent feature. The detected glucuronic acid indicates the use of glucose as an energy source by AAB. The AAB species, that is characterized by a high ability to synthesize gluconic acid and a low ability to synthesize acetic acid is *Gluconobacter oxydans.* The obtained results may indicate its main share in the microbiological composition of the developed beverages [6,64,67].

## 5. Conclusions

As a result of the research, drinks of high microbiological, sensory and physicochemical quality were obtained. What is more, the developed fermented milk drinks have a potential health-promoting value, thanks to the content of active microflora and organic acids, which have a confirmed positive effect on the human body. From a technological point of view, the most suitable samples for industrial production are those that contain the lowest proportion of milk inoculum in the composition and the time of preparation is shorter. However, the developed products require further optimization to shorten the milk inoculum and fermented milk drink elaboration, which will allow reducing the production costs of this type of beverage. One of the decisive elements in obtaining a microbiologically stable fermentation inoculum will be to determine the proportion of individual SCOBY microorganisms in the milk environment. An important element of the optimization will be also the sensory characteristics of the products and further chemical research to deepen the knowledge on the content of other nutrients.

## Figures and Tables

**Figure 1 microorganisms-09-00123-f001:**
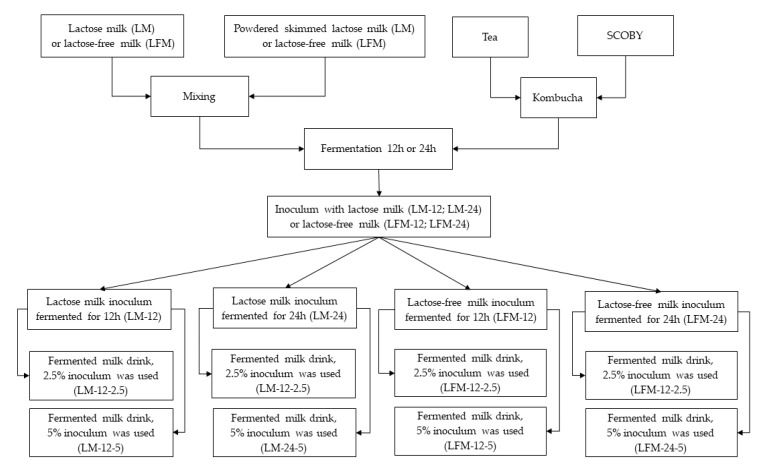
Schematic diagram of the experiment and tested variants.

**Figure 2 microorganisms-09-00123-f002:**
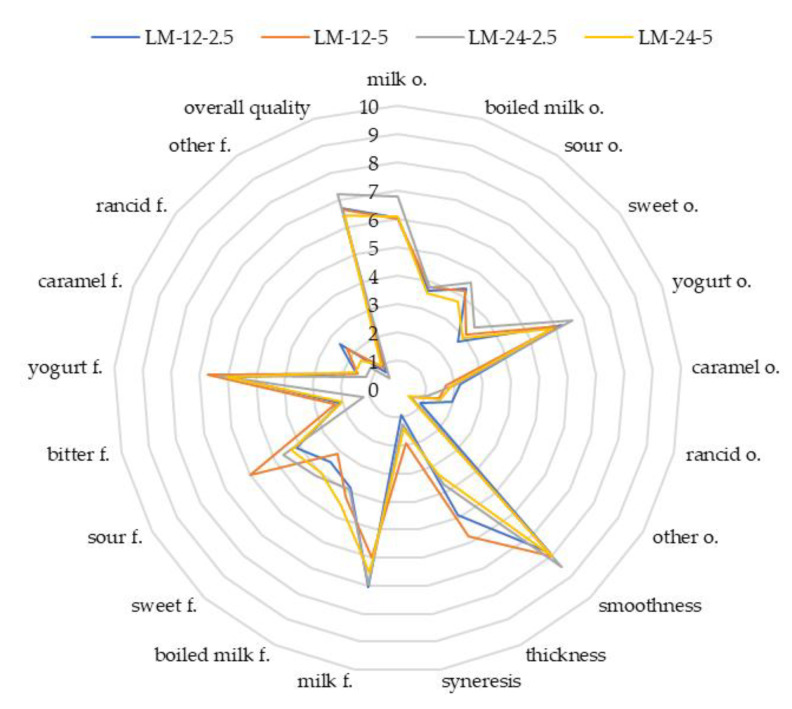
The sensory analysis of fermented lactose milk drinks. Explanations: conventional unit (0–10 c.u.); f.—flavour; o.—odour; LM-12-2.5—fermented milk drink with 2.5% (*v*/*v*) of a 12-h inoculum; LM-12-5—fermented milk drink with 5% (*v*/*v*) of a 12-h inoculum; LM-24-2.5—fermented milk drink with 2.5% (*v*/*v*) of a 24-h inoculum; LM-24-5—fermented milk drink with 5% (*v*/*v*) of a 24-h inoculum; n = 14.

**Figure 3 microorganisms-09-00123-f003:**
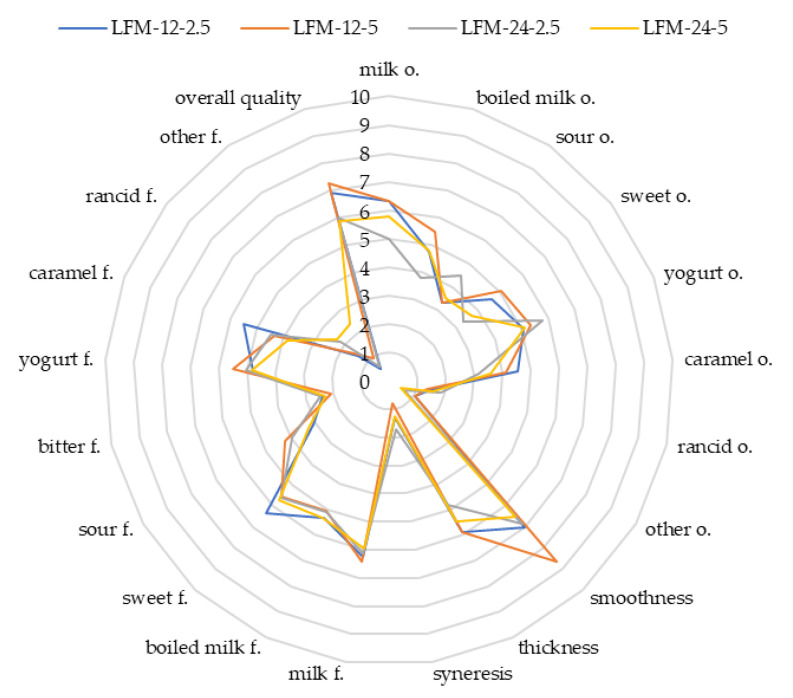
The sensory analysis of fermented lactose-free milk drinks. Explanations: conventional unit (0–10 c.u.); f—flavour; o—odour; LFM-12-2.5 -fermented lactose-free milk drink with 2.5% (*v*/*v*) of a 12-h inoculum; LFM-12-5—fermented lactose-free milk drink with 5% (*v*/*v*) of a 12-h inoculum; LFM-24-2.5—fermented lactose-free milk drink with 2.5% (*v*/*v*) of a 24-h inoculum; LFM-24-5—fermented lactose-free milk drink with 5% (*v*/*v*) of a 24-h inoculum; n = 14.

**Table 1 microorganisms-09-00123-t001:** The mean number of the selected microbial species in the milk inoculum and its pH value.

Inoculum Type	Microbial Species (log CFU mL^−1^)				pH Value
	LAB	BB	AAB	Yeast	
LM-12	9.23 ± 0.67 ^a^	7.46 ± 0.31 ^a^	9.20 ± 0.45 ^a^	5.58 ± 0.04 ^a^	4.67 ± 0.02 ^ac^
LM-24	8.94 ± 1.01 ^a^	7.28 ± 0.23 ^a^	7.90 ± 0.60 ^b^	5.30 ± 0.08 ^b^	4.28 ± 0.04 ^b^
LFM-12	9.19 ± 0.11 ^a^	7.92 ± 0.37 ^a^	9.01 ± 0.07 ^a^	5.99 ± 0.43 ^abc^	4.67 ± 0.03 ^ac^
LFM-24	10.23 ± 0.13 ^a^	7.40 ± 0.29 ^a^	8.70 ± 0.95 ^a^	6.58 ± 0.48 ^c^	4.38 ± 0.03 ^d^

Explanation: LM-12—lactose milk inoculum fermented for 12 h; LM-24—lactose milk inoculum fermented for 24 h; LFM-12—lactose-free milk inoculum fermented for 12 h; LFM-24—lactose-free milk inoculum fermented for 24 h; LAB—lactic acid bacteria; BB—bifidobacteria; AAB—acetic acid bacteria; means followed by different letters ^a^, ^b^, ^c^, ^d^ are significantly different (*p* < 0.05); (n = 3).

**Table 2 microorganisms-09-00123-t002:** The mean number of the selected microbial species in the designed fermented milk beverages and the pH value.

Product Type	Microbial Species (log CFU mL^−1^)				pH Value
	LAB	BB	AAB	Yeast	
LM-12-2.5	9.67 ± 0.35 ^a^	7.01 ± 0.23 ^a^	9.32 ± 0.78 ^a^	6.37 ± 1.86 ^a^	4.66 ± 0.05 ^a^
LM-12-5	9.20 ± 0.15 ^a^	6.71 ± 0.38 ^a^	9.23 ± 0.24 ^a^	4.09 ± 0.08 ^a^	4.53 ± 0.05 ^b^
LM-24-2.5	9.32 ± 0.23 ^a^	7.08 ± 0.09 ^a^	8.34 ± 0.55 ^b^	4.26 ± 0.05 ^a^	4.70 ± 0.04 ^a^
LM-24-5	9.50 ± 0.52 ^a^	7.13 ± 0.10 ^a^	8.77 ± 0.11 ^ab^	5.31 ± 1.52 ^a^	4.68 ± 0.04 ^a^
LFM-12-2.5	9.32 ± 0.30 ^a^	8.22 ± 0.84 ^b^	8.76 ± 0.16 ^ab^	4.91 ± 1.36 ^a^	4.65 ± 0.06 ^a^
LFM-12-5	9.28 ± 0.04 ^a^	6.85 ± 0.15 ^a^	8.83 ± 0.05 ^ab^	5.12 ± 1.49 ^a^	4.56 ± 0.01 ^b^
LFM-24-2.5	9.08 ± 0.21 ^a^	7.86 ± 0.71 ^c^	9.14 ± 0.58 ^ab^	5.40 ± 2.00 ^a^	4.71 ± 0.04 ^a^
LFM-24-5	9.67 ± 0.69 ^a^	7.79 ± 0.88 ^c^	9.45 ± 0.66 ^a^	6.35 ± 1.87 ^a^	4.70 ± 0.03 ^a^

Explanation: LM-12-2.5—fermented milk drink with 2.5% (*v*/*v*) of a 12-h inoculum; LM-12-5—fermented milk drink with 5% (*v*/*v*) of a 12-h inoculum; LM-24-2.5—fermented milk drink with 2.5% (*v*/*v*) of a 24-h inoculum; LM-24-5—fermented milk drink with 5% (*v*/*v*) of a 24-h inoculum; LFM-12-2.5—fermented lactose-free milk drink with 2.5% (*v*/*v*) of a 12-h inoculum; LFM-12-5—fermented lactose-free milk drink with 5% (*v*/*v*) of a 12-h inoculum; LFM-24-2.5—fermented lactose-free milk drink with 2.5% (*v*/*v*) of a 24-h inoculum; LFM-24-5—fermented lactose-free milk drink with 5% (*v*/*v*) of a 24-h inoculum; LAB—lactic acid bacteria; BB—bifidobacteria; AAB—acetic acid bacteria; means followed by different letters ^a^, ^b^, ^c^ are significantly different (*p* < 0.05); (n = 3).

**Table 3 microorganisms-09-00123-t003:** The content of sugars, vitamin C and organic acids in the designed products and the mixture of milk and powdered milk.

Product Type	Content (g/100 g)			Vitamin C Content (mg/100 g)	Organic Acid Content (g/100 g)					
	Lactose	Glucose	Galactose		Lactic Acid	Citric Acid	Orotic Acid	Glucuronic Acid	Pyruvic Acid	Acetic Acid
LM-C	6.53 ± 0.04 ^a^	nd	0.09 ± 0.01 ^a^	0.33 ± 0.03 ^a^	0.01 ± 0.00 ^a^	0.15 ± 0.01 ^a^	0.08 ± 0.01 ^a^	nd	0.05 ± 0.00 ^a^	nd
LM-12-2.5	4.83 ± 0.07 ^b^	nd	0.79 ± 0.03 ^b^	0.30 ± 0.02 ^a^	0.70 ± 0.02 ^b^	0.15 ± 0.01 ^a^	0.07 ± 0.01 ^a^	0.04 ± 0.00 ^a^	0.02 ± 0.00 ^b^	nd
LM-12-5	4.76 ± 0.04 ^c^	nd	0.84 ± 0.04 ^c^	0.30 ± 0.02 ^a^	0.70 ± 0.02 ^b^	0.15 ± 0.01 ^a^	0.08 ± 0.01 ^a^	0.03 ± 0.00 ^b^	0.02 ± 0.00 ^b^	nd
LM-24-2.5	3.74 ± 0.04 ^d^	nd	1.25 ± 0.04 ^d^	0.32 ± 0.02 ^a^	0.77 ± 0.02 ^c^	0.14 ± 0.01 ^a^	0.08 ± 0.01 ^a^	0.04 ± 0.00 ^c^	0.02 ± 0.00 ^b^	nd
LM-24-5	3.69 ± 0.07 ^e^	nd	1.18 ± 0.02 ^e^	0.29 ± 0.01 ^a^	0.76 ± 0.02 ^c^	0.14 ± 0.01 ^a^	0.07 ± 0.01 ^a^	0.04 ± 0.00 ^d^	0.02 ± 0.00 ^b^	nd
LFM-C	nd	3.27 ± 0.02 ^a^	4.28 ± 0.03 ^f^	0.38 ± 0.03 ^a^	0.01 ± 0.00 ^a^	0.18 ± 0.01 ^a^	0.10 ± 0.01 ^a^	nd	0.03 ± 0.00 ^a^	nd
LFM-12-2.5	nd	2.46 ± 0.05 ^b^	4.29 ± 0.02 ^f^	0.35 ± 0.02 ^a^	0.58 ± 0.01 ^b^	0.17 ± 0.01 ^a^	0.09 ± 0.01 ^a^	0.03 ± 0.00 ^e^	0.02 ± 0.00 ^b^	nd
LFM-12-5	nd	2.41 ± 0.04 ^c^	4.24 ± 0.05 ^f^	0.35 ± 0.03 ^a^	0.60 ± 0.01 ^b^	0.17 ± 0.01 ^a^	0.09 ± 0.01 ^a^	0.03 ± 0.00 ^f^	0.02 ± 0.00 ^b^	nd
LFM-24-2.5	nd	2.18 ± 0.03 ^d^	4.03 ± 0.07 ^g^	0.36 ± 0.02 ^a^	0.66 ± 0.01 ^c^	0.16 ± 0.01 ^a^	0.09 ± 0.01 ^a^	0.04 ± 0.00 ^g^	0.02 ± 0.00 ^b^	nd
LFM-24-5	nd	2.02 ± 0.04 ^e^	3.97 ± 0.07 ^h^	0.35 ± 0.02 ^a^	0.67 ± 0.01 ^c^	0.16 ± 0.01 ^a^	0.09 ± 0.00 ^a^	0.04 ± 0.00 ^h^	0.02 ± 0.00 ^b^	nd

Explanation: LM-C—control sample of non-fermented lactose milk; LFM-C—control sample of non-fermented lactose-free milk; LM-12-2.5—fermented milk drink with 2.5% (*v*/*v*) of a 12-h inoculum; LM-12-5—fermented milk drink with 5% (*v*/*v*) of a 12-h inoculum; LM-24-2.5—fermented milk drink with 2.5% (*v*/*v*) of a 24-h inoculum; LM-24-5—fermented milk drink with 5% (*v*/*v*) of a 24-h inoculum; LFM-12-2.5—fermented lactose-free milk drink with 2.5% (*v*/*v*) of a 12-h inoculum; LFM-12-5—fermented lactose-free milk drink with 5% (*v*/*v*) of a 12-h inoculum; LFM-24-2.5—fermented lactose-free milk drink with 2.5% (*v*/*v*) of a 24-h inoculum; LFM-24-5—fermented lactose-free milk drink with 5% (*v*/*v*) of a 24-h inoculum; nd—not detected; means followed by different letters ^a^, ^b^, ^c^, ^d^, ^e^, ^f^, ^g^, ^h^ are significantly different (*p* < 0.05); (n = 3).

## Data Availability

No new data were created or analyzed in this study. Data sharing is not applicable to this article.

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
