# Peer review of "Application of the “SCOBY” and Kombucha Tea for the Production of Fermented Milk Drinks"

_microorganisms, 2021, doi:10.3390/microorganisms9010123_

Round 1

Reviewer 1 Report

Comments to the Author;

General comments:

The manuscript mainly assessed the possibility of using the Kombucha and SCOBY as starter culture for fermented milk drinks productions. The study is interesting and relatively novel. However, there are some shortcomings listed as follows:

Abstract

Line 20: There were no physical properties shown in this manuscript, please change “physicochemical methods” to “Chemical methods”.

Line 21: Change “Aa” for “As”.

Lines 22-24: The abstract was used to display what was done in the research, but numbers of sugars and organic acid profile in yoghurts or kefirs were not investigated in this study. Please improve this sentence.

Introduction

The introduction needs to explain why authors attempted to use lactose-free variant and the traditional milk to produce a new fermented milk drink. And explain why you choose microbiological, chemical and sensory properties to do your research?

Materials and Methods

Authors declare to use lactose-free variant and the traditional milk in this study, however, it is hard to find them in the section of “Materials and Methods”. Please highlighted it.

Line 97: Please show the detail information about the SCOBY starter culture, for example, the name of species in it.

Lines 101-119: It is hard to understand the technological process and the abbreviation of each sample’s name. Please show them in a flow chart.

Lines 121-137: Please move this part to the front of 2.2.5 statistical analysis. Sensory properties were discussed at last.

Results

What’s the meaning of “LM” and “LFM” in Table 1 and Table 2? Why you do not use “LM12, LM24, LFM12 and LFM24”? In addition, these two tables could be merged into one.

Line 229: “9 log CFU ml−1”, format mistake, please fix them all.

Lines 238, 264, 292: Please label significance in the Tables.

Discussion

Lines 353-364: Authors try to compare the different of microbiological quality in samples with literature data. However, the microbial composition of traditional yoghurt starter culture, kefir grains and SCOBY was apparently different. So, these sentences were meaningless.

Reviewer 2 Report

A very interesting manuscript on the use of SCOBY for the production of a fermented milk beverage. The manuscript needs several improvements:

l. 35-37. please rephrase to improve clarity

l. 54-58. the ability to function as a probiotic culture is strain-dependent, not species genera or type. These lines are scientifically inaccurate and have to be accordingly revised and enriched with appropriate citations.

l. 73. references 20 and 21 are not relevant to the topic of this line. Please use relevant references, especially ones that mention occurrence of bifidobacteria in SCOBY

l. 109, l. 118 please explain more the types of inoculum employed and the types of fermented milk beverages obtained.

l. 126. it should read ‘…the following characteristics were assessed:’

throughout the text. please pay attention to the use of English language, especially the use of the appropriate number and tense, e.g. l. 175 (remove ‘a’), l. 198 (replace ‘samples’ with ‘sample’), l. 176 (replace ‘triplicated’ with ‘triplicate’), l. 209 (replace ‘injection’ with ‘injected’). In addition, ‘-1’ should be superscripted (e.g. l. 181, 229, 230).

l.236, legends of tables 1 & 2. The term ‘microbiological quality’ cannot be used in these cases because the population of only LAB, BB, AAB & yeasts was assessed. Assessment of the microbiological quality of a sample includes also the detection of spoilage and pathogenic microorganisms.

please merge paragraphs 3.1 with 3.3 and 3.2 with 3.4

Tables 1 & 2 & 3. ANOVA and proper annotation (per columns) will be very helpful and will assist the reader

l. 302 please replace ‘statistical differences’ with ‘statistically significant differences’

l.449-453 and throughout the text. Which acid did the authors studied, gluconic or glucuronic?

table 3, l. 454-457. acetic acid bacteria may oxidize a wide range of carbohydrates as well as ethanol and lactic acid. In table 2, the AAB population is reported at 9 log CFU/mL; however, no acetic acid was detected. This is an impressive result that has not been properly explained

please pay attention to the format of the references, e.g. 17: in capital, 9: capitalized first letter, 1: no capitals, scientific names should be written in italics (e.g. 12), reference 34 is the same as 56.

Round 2

Reviewer 2 Report

The authors have successfully addressed the comments

This manuscript is a resubmission of an earlier submission. The following is a list of the peer review reports and author responses from that submission.